# Study on the Deformation Behavior of Two Phases during the Low Cycle Fatigue of UNS S32750 Duplex Stainless Steel

**DOI:** 10.3390/ma17143390

**Published:** 2024-07-09

**Authors:** Shun Bao, Han Feng, Zhigang Song, Jianguo He, Xiaohan Wu, Yang Gu

**Affiliations:** Special Steel Research Institute of General Iron and Steel Research Institute Co., Ltd., Beijing 100081, China; baoshun1994@163.com (S.B.); fenghan@nercast.com (H.F.); hejianguo@nercast.com (J.H.); wuxiaohan@nercast.com (X.W.); thiagoyoungkoo@163.com (Y.G.)

**Keywords:** S32750 duplex stainless steel, low cycle fatigue, plastic deformation, dislocation evolution

## Abstract

In this paper, the deformation behavior of UNS S32750 (S32750) duplex stainless steel during low cycle fatigue was studied by controlling the number of cycles. The microstructure of the specimens under different cycles was characterized by optical microscope (OM), scanning electron microscope (SEM), electron backscatter diffraction (EBSD), and transmission electron microscope (TEM). The microhardness of the two phases was measured by a digital microhardness instrument. The results showed that the microhardness of ferrite increases significantly after the first 4000 cycles, while the austenite shows a higher strain hardening rate after fatigue fracture, and the microhardness of ferrite and austenite increases by 23 HV and 87 HV, respectively. The two-phase kernel average misorientation (KAM) diagram showed that the continuous accumulation of plastic deformation easily leads to the initiation of cracks inside the austenite and at the phase boundaries. The evolution of dislocation morphology in the two phases was obviously different. With the increase in cycle number, the dislocation in ferrite gradually transforms from dislocation bundles and a dislocation array to a sub-grain structure, while the dislocation in austenite gradually develops from dipole array to an ordered Taylor lattice network structure.

## 1. Introduction

Duplex stainless steel (DSS) is a kind of stainless steel with a similar volume fraction of α phase and γ phase, which combines good corrosion resistance and excellent mechanical properties. Compared with austenitic stainless steel, DSS has higher strength and elongation [1], so it is widely used in construction [2], offshore oil and gas exploitation [3], chemical processing, and the nuclear industry [4,5,6]. However, many components are required to serve under long-term alternating load conditions, and the continuous accumulation of fatigue damage can lead to fatigue failure and accidents. Therefore, to ensure the safe service of components and improve service life, the fatigue performance of duplex stainless steel has always been the focus of international research.

The difference in the properties of ferrite and austenite in S32750 DSS, as well as the existence of phase boundaries and grain boundaries, means that the plastic deformation of the two phases is dynamically coordinated under external loads. Therefore, the response behavior of the test steel under cyclic loading is significantly different from that of ordinary single-crystal materials [7]. The deformation behavior of the two phases affects the initiation and propagation of fatigue cracks during fatigue loading. Zhang et al. [8] used neutron diffraction and EBSD techniques to study the deformation and orientation evolution of the two phases in 2205 duplex stainless steel from the initial to the fracture state. The results indicated that the orientation of ferrite gradually changes from <001> to <111>, while the orientation of austenite is more homogeneous and the texture does not obviously change. Polak et al. [9,10] compared the low-cycle fatigue behavior of Sancro 25 steel at room temperature and high temperatures and found that the dislocation morphology of austenite at two temperatures showed different characteristics. At room temperature, the single slip is dominant in austenite, while at high temperatures, the dislocation bands formed by two slip systems are more evenly cross-distributed. This high density of dislocation bands makes the primary dislocation movement more difficult and reduces the fatigue life. Zhu et al. studied the low cycle fatigue behavior of 2507 duplex stainless steel at a constant strain amplitude of ±1%. The results show obvious strain anisotropy in the austenitic phase, which also causes the steel to harden in the first 10 cycles, and the dislocation annihilation and rearrangement of the two phases cause the steel to soften. This uneven distribution of stress between lattices also affects the formation of sub-structures during fatigue processes [11]. Dönges et al. [12] investigated the crack initiation mechanism of high-cycle and ultra-high-cycle fatigue in 2205 duplex stainless steel and found that the inclination angle and torsion angle between the activated slip plane in austenite grains and the slip plane of adjacent ferrite grains (which has the largest Schmid factor) were small enough that the dislocation could transfer to the harder ferrite phase. This transition is local and often accompanied by the initiation of microcracks. Fu et al. [13] studied the effect of residual stress and strain on the very-high-cycle fatigue properties of AISI 318LN duplex stainless steel. The results show that the residual strain changes from ferrite to austenite with the increase in cycle times. This transfer occurs when the γ/α phase boundary is packed with enough dislocations, and the strain can be easily transferred at the γ/γ phase boundary. The interaction between the two phases is the main mechanism of microcrack initiation in DSS. It can be seen that the initiation of DSS fatigue cracks is closely related to two-phase hardness, two-phase dislocation accumulation, and strain distribution. However, there is still a lack of research on the relationship between the two-phase deformation behavior and fatigue crack initiation, and the dislocation evolution of the two phases during deformation. Therefore, a series of studies on the deformation behavior of S32750 DSS is helpful to understand the formation and propagation of microcracks in the fatigue process of the steel, and provides a theoretical basis for improving its fatigue performance.

## 2. Experimental Materials and Methods

### 2.1. Preparation of Test Steel

The test steel is a Φ180 × 200 mm (height) ingot provided by Zhejiang Jiuli Special Materials Technology Co., Ltd. (Jiuli, China), which was forged by Hebei sanhe Forging Plant. The size of the formed slab is 245 × 115 × 45 mm. In order to ensure the subsequent smooth rolling, the plate is uniformly polished up and down along the thickness direction of the plate until the final plate thickness is 40 mm. The slab was heated to 1200 °C for 2 h, and then 17%, 20%, and 25% three passes of rolling treatment were carried out to ensure that the final rolling temperature was no less than 1000 °C. The chemical composition of the test steel is shown in Table 1.

In order to ensure that the matrix does not contain harmful precipitates [14], the solution treatment temperature of S32750 duplex stainless steel is usually higher than 1050 °C. In order to make the content of α and γ phases close to 1:1, the heat treatment process of the test steel is 1080 °C, holding for 40 min before water-cooling.

### 2.2. Tensile and Fatigue Testing

The tensile test was carried out using the GB/T 228.1-2021 [15]. The experimental results show that the yield and tensile strength of S32750 DSS after solution treatment at 1080 °C can reach 560 MPa and 825 MPa, and the elongation and reduction in area are 35% and 66%. The low cycle fatigue performance is completed according to the GB/T 3075-2021 [16]. The fatigue trial was carried out at room temperature, the maximum stress was 600 MPa, R = −1, the frequency was 1 Hz, and the loading waveform was sine wave. The rod-like fatigue specimen is shown in Figure 1, and the surface roughness Ra is less than 0.2 μm. Both samples were prepared transverse along the plate. In order to explore the deformation behavior of the two phases in the fatigue process, five samples were subjected to fatigue experiments of 2000, 4000, 6000, 8000, and fracture state. The experimental parameters are shown in Table 2.

### 2.3. Microstructure Observation and Microscopic Characterization

The ferrite phase was colored by 15% KOH solution constant-voltage electrolysis method, with a voltage of 5 V and time of 10~15 s, for phase ratio statistics. Using the 10% H_2_SO_4_ + 1 g KMnO_4_ solution water bath method, at a temperature of 50 °C with an immersion time of 3 h, α, γ grain boundaries were colored, and then the Nano Measure software (1.2.0) was used to calculate the grain size. The microhardness of the two phases was measured by an FM-300 digital microhardness tester, and the loading load was 0.05 kgf. Ten positions were selected for both phases to measure and take the average. EBSD samples were prepared by constant-voltage electrolysis of 10% HClO_4_ solution at 25 V for 30 s. TEM thin disks were prepared via constant-current electrolysis in 10% HClO_4_ solution, with a current of 75 mA and pore size of 80. The database used for Thermo-calc calculation is TCFE12: Steel/Fe-Alloys v12.0.

## 3. Result

### 3.1. Phase Ratio and Grain Size

The content of the alloying elements in S32750 DSS was more than 35%, which led to an increase in the precipitation tendency of harmful phases, such as the M_23_C_6_, σ phase, and χ phase. Through Thermo calc thermodynamic calculation software (2023.1.111866-468), the precipitation temperature, content of each phase, and the content of components in the phase can be effectively known. The main composition of the test steel was used as the input value in thermodynamic calculations, and the relationship between each phase and temperature is shown in Figure 2. It can be seen from the figure that when the temperature exceeds 1030 °C, the matrix contains only ferrite and austenite phases. When the temperature is close to 1080 °C, the volume fraction of the two phases is closest to 1:1.

The two-phase microstructure morphology of steel is shown in Figure 3a. The black is the α phase, and the white island structure is the γ phase. After hot rolling and solid solution treatment, the two-phase structure in the matrix is elongated and evenly distributed in layers. According to statistics, the proportion of α and γ phases is 51.3% and 48.7%. The two-phase grain boundaries and grain morphology are shown in Figure 3b. The grain size data of ferrite and austenite are shown in Figure 3c,d. It is observed that ferrite and austenite are still dominated by grains of less than 50 μm, with a maximum size of 250 μm. The austenite grain size is closer to the normal distribution, and the maximum size is 120 μm.

### 3.2. Fatigue Crack Source and Fracture Morphology

For high-strength steel, inclusions can easily cause stress concentration and fatigue fracture during cycling, as the steel is more sensitive to the size of the inclusions [17]. In contrast, DSS with lower strength is less sensitive to inclusion size. The low cycle fatigue fracture morphology and fatigue fracture diagram of S32750 DSS under the stress amplitude of 600 MPa are shown in Figure 4. (a) is the macroscopic morphology of the fracture at low magnification, (b) and (c) are the enlarged diagram of the crack initiation source, and (d) is the surface crack initiation diagram of the rod fatigue specimen. After observing the macroscopic fracture morphology, no oxide inclusions were found on the surface and sub-surface, but two intrusion and extrusion traces were found on the surface edge, as shown in Figure 4b,c. Under a continuous load, plastic deformation accumulates and passes to the surface, causing relative displacement on the surface, and gradually evolves into the crack source.

The fatigue fracture is divided into four parts, including crack propagation zone I and II, static rupture zone III, and instantaneous fracture zone IV. Clear boundaries exist between crack propagation zones I and II and static rupture zone III, and the morphology on both sides of the boundary is also different. This may be due to the different crack propagation speeds during the fatigue fracture process, forming different cleavage and ductile fracture morphologies. Figure 5a–c correspond to the local enlargements of the I, II, and III positions in Figure 4, respectively. Region I is the cleavage fracture, which is composed of many cleavage planes, secondary cracks (as shown by red arrows in Figure 5a), and a small amount of fatigue striations, as shown in Figure 5a, and the crack propagation in this region is extremely fast. Figure 5b is the typical morphology of region II. A large number of fatigue striations occupy almost the whole field of view. In this region, the fatigue crack can expand more stably. Under tensile stress, plastic deformation causes the crack tip to become passivated. When unloading, the stress is reduced, which can make the crack tip become sharp again, and finally a large number of fatigue striations [18] are formed on this area. In the later periods of the crack propagation, as shown in Figure 5c, massive dimples of different sizes are formed. The formation of regions I and II consumed a lot of energy. The crack experienced the rapid expansion of zones I and II, which consumed a lot of energy, and the crack growth rate was slowed down, making it easy for a dimple morphology to form.

### 3.3. Change in Two Phases of Microhardness under Different Cycles

The two-phase crystal structure in the S32750 DSS matrix is different, so its ability to withstand deformation is also inconsistent. The macroscopic plastic deformation of materials is the result of the continuous accumulation of microscopic deformation. Local stress and strain concentration will cause crack initiation. The crack initiation of DSS is a complicated phenomenon affected by many factors [19]. Internal factors such as the elastic–plastic difference of ferrite and austenite, and the change in elements, external factors including experimental parameters such as stress amplitude and frequency [20,21] will affect the fatigue deformation mechanism and crack initiation. Therefore, studying the deformation behavior of the two phases during the cyclic process is of great help in elucidating the fatigue failure mechanism of DSS. The microhardness can reflect the degree of deformation in grains of α and γ phases. The hardness increment in ferrite and austenite in five specimens with different cycles is shown in Figure 6.

The initial microhardness of ferrite is higher than that of austenite after the solution treatment. Comparing (a) and (b) in Figure 6, it is found that the microhardness increments of α and γ phases show completely different trends. After 2000 cycles, the microhardness of the two phases increased slightly; in the early stage of the cycle, the deformation mainly occurred in the soft phase (austenite) and the planar slip is relatively easy, making it difficult for strain to accumulate. As the number of cycles increases to 4000, the ferrite is obviously hardened, the microhardness increases by 20 HV, and the austenite hardness increases by 9 HV, indicating that a large amount of deformation is accumulated in ferrite during this process. Subsequently, when the number of cycles increases to 8000, the microhardness in the ferrite grains continues to increase, while the hardness in the austenite grains also shows a significant growth trend. The specimen fractured after 9387 cycles, and the microhardness of the two phases near the fracture showed that the microhardness increment in ferrite was 23 HV, while the microhardness increment in austenite was 87 HV, and the strain hardening was much higher than that of ferrite.

## 4. Discussion

### 4.1. Evolution Rule of Two-Phase KAM in the Cyclic Process

KAM mappings are usually used to characterize the degree of plastic deformation in the test area. Figure 7 shows the high- and low-angle grain boundaries and the corresponding KAM distribution of the test steel from the initial state to 8000 cycles. The brighter green color in KAM indicates greater cumulative plastic deformation in this region. It can be seen that the plastic deformation in the initial state is mainly concentrated in the high-angle grain boundaries of austenite (as shown by white arrows and white circles), two-phase phase boundaries (as shown by black arrow and black box), and low-angle grain boundaries of ferrite (as shown by yellow arrows and yellow circles). With the increase in cycle times to 4000, it is obvious that plastic deformation accumulates at the low-angle grain boundaries in the ferrite (as shown by yellow arrows and yellow circles). In the KAM diagram of 6000 cycles, the color of the phase boundary and some austenite grains is brighter, indicating that plastic deformation is beginning to accumulate in the austenite(as shown by black box). As the number of cycles increases to 8000, the regions with large plastic deformation mainly include phase boundaries, the austenite interior(as shown by black box), and low-angle boundaries (as shown by yellow arrows and yellow circles).

The low- and high-angle grain boundaries can reflect the movement, proliferation, and entanglement of dislocations to some extent. Figure 8 shows the change in the proportion of grain boundaries of ferrite and austenite from the initial to the fracture state. It can be seen that the grain boundary ratio of ferrite and austenite shows a completely different trend with the increase in the number of cycles. The proportion of low-angle grain boundaries in ferrite increases to 60% when the number of cycles reaches 2000, and changes slightly when the number of cycles reaches 8000. After fracture, the proportion of the low-angle grain boundary of ferrite is close to 100%. The proportion of the low-angle grain boundary in austenite was almost the same before 8000 cycles. After fatigue fracture, the number of small-angle grain boundaries exceeded 50%. In the low-cycle fatigue process of UNS S32750 duplex stainless steel, the ferrite deforms in a way that forms a low-angle grain boundary, while the austenite forms a low-angle grain boundary only when fracture occurs.

The fatigue fracture of the tested steel occurred after 9387 cycles, and the two-phase information of the longitudinal parabolic surface of the fracture is shown in Figure 9. A large number of low-angle grain boundaries were formed near the fracture. It can be seen from the KAM diagram that a large amount of deformation accumulated in the ferrite, austenite, and two-phase phase boundary. The ferrite grain 1 near the edge of the fracture fragmented and a large gap was formed. Austenite grains 2, 3, 4, and 5 not only form a gap at the phase boundary due to the large plastic deformation, but also show a large amount of deformation inside the grain, resulting in grain fragmentation and splitting. The angle between the splitting directions of austenite grains 3 and 4 and the loading direction is about 45°. Austenite is more likely to cause intragranular cracking than ferrite; therefore, after long-term loading, cracks easily initiate inside the austenite and the phase boundary. By observing the grain orientation distribution, it is found that the ferrite tends to have a <111>∥RD (roll direction) texture, and there are also orientation differences within the grains, indicating that the long-term dislocation accumulation in the ferrite can easily cause the grains to rotate. Austenite tends to have <101>∥RD texture, and only a small amount of grains rotated.

### 4.2. Relationship between Dislocation Structure and the Number of Cycles

The evolution of a dislocation structure during cycling at nanoscale can be observed by TEM. The generation and development of dislocations are related to the overall plastic deformation. Therefore, in order to reveal the essence of fatigue fracture, it is necessary to combine macro- and micro methods. Figure 10 shows the internal dislocation morphology of ferrite grains from the initial state of the test steel to 8000 cycles. By observing (a–c) in Figure 10, a certain number of dislocations can be observed in the initial state of the test steel, and they show three different morphologies. The first is a dislocation bundle with a certain thickness of 27~200 nm formed by dislocation stacking entanglement, and this dislocation bundle presents a certain hindrance to the new dislocation. The second is a parallel and straight dislocation array [22] in two directions. The third type is the nearly uniform dislocation dipole [23] formed by the screw dislocation motion. Figure 10d–f shows the dislocation morphologies in ferrite after 2000 cycles, which does not change much and is still dominated by the dislocation bundle and dislocation dipole. When the number of cycles increases to 4000, as shown in Figure 10g–i, the density of parallel dislocation arrays increases significantly, forming a grid morphology. Figure 10j–l shows the dislocation morphology of ferrite after 6000 cycles. It is found that, in addition to the continuous increase in the density of the dislocation array, sub-grain boundaries are formed inside the grains. When the cycles increase to 8000, as shown in Figure 10m–o, the number of sub-grain boundaries begins to increase, and dislocations continue to accumulate (as shown by red circle in Figure 10o) at the two-phase phase boundary.

Figure 11 shows the internal dislocation morphology of austenite from the initial state to 8000 cycles. Among them, (a) and (b) are the internal dislocation morphologies of austenite in the initial state, which are mainly composed of dipole arrays [24] with a spacing of approximately 130nm. The dipole arrays have been bent many times, indicating that after solution treatment, there are still deformation traces caused by forging and hot rolling in austenite. Figure 11b shows the cross-entanglement of the dipole array. After 2000 cycles, the dislocation morphology in austenite changed significantly, as shown in Figure 11c,d. Due to the continuous tension and compression loading, the movement of multiple slip systems is intensified, resulting in severe deformation of the dipole array, increased dislocation entanglement, and an ordered network-like dislocation Taylor lattice structure [25] being formed inside the grains. It can be seen that a large amount of deformation occurred in austenite at the early stage of the fatigue cycle. Figure 11e,f show the dislocation morphology of austenite after 4000 cycles, which is still dominated by a regular grid morphology. When the number of cycles increases to 6000 and 8000 times, the dislocation morphology of austenite is shown as (g) and (h) in Figure 11, and the internal dislocation tangles (as shown by yellow circle in Figure 11g) are further aggravated. At the same time, the phase boundary, a barrier that effectively hinders the dislocation movement, also accumulates a large number of dislocations.

### 4.3. Fatigue Crack Initiation Process

During the fatigue process, the dislocation proliferation and development of the two phases show different trends, which affects the accumulation of plastic deformation and the location of crack initiation. The development process of the two-phase dislocation and the initiation of fatigue cracks are shown in Figure 12. Dislocation tangles in ferrite grains are increasing, and a large number of dislocations remain inside the grains. The increasing density of dislocation arrays will eventually evolve into sub-grain boundaries. However, the dislocations in austenite are mainly the continuous increase in dislocation dipoles. Finally, the network Taylor lattice structure is formed and most of the dislocations are accumulated in several slip systems and accumulated at the phase boundary of the two phases. Therefore, fatigue cracks are more likely to initiate at the two-phase boundaries and within the austenite grains.

## 5. Conclusions and Prospects

In this paper, the two-phase deformation behavior of UNS S32750 duplex stainless steel in multiple states, both before fatigue fracture and under fatigue fracture, was studied. The strain accumulation and dislocation evolution of ferrite and austenite were studied by EBSD, TEM methods, etc. The main results are summarized as follows:(1)Based on the microhardness test results, in the first 2000 cycles, the microhardness of the two phases of S32750 duplex stainless steel is slightly increased. When the number of cycles increased to 4000 times, the microhardness of ferrite increased significantly, indicating that a large amount of plastic deformation was accumulated in ferritic grains. When a sample faces fatigue fracture, the microhardness of ferrite and austenite increased by 23 and 87, respectively, and the strain-hardening rate of austenite was much higher than that of ferrite.(2)After 9387 cycles, fatigue fracture occurred. Near the fracture, the austenite grains are more likely to be fragmented, and the phase boundaries are also easy to form gaps. Therefore, microcracks easily initiate in austenite grains and phase boundaries. After fatigue fracture, ferrite grains tend to form a <111>∥RD texture, and austenite tends to form a <101>∥RD texture.(3)With the increase in cycle numbers, the development of a dislocation morphology in ferrite and austenite shows different trends. The initial dislocation beam and dislocation array in ferrite gradually change into a sub-grain boundary structure. The initial dipole array in austenite gradually deforms and tangles, and finally forms a regular network Taylor lattice structure.

The two-phase plastic deformation and dislocation evolution under different cycles were systematically studied. Fatigue crack initiation was not observed. In the future, we hope to observe the generation of fatigue cracks in UNS S32750 duplex stainless steel through in situ fatigue experiments.

## Figures and Tables

**Figure 1 materials-17-03390-f001:**
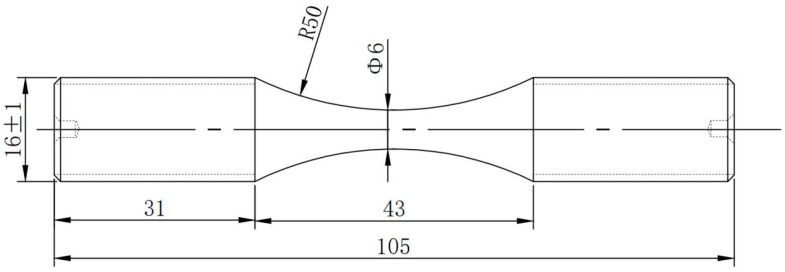
Rod fatigue specimen size.

**Figure 2 materials-17-03390-f002:**
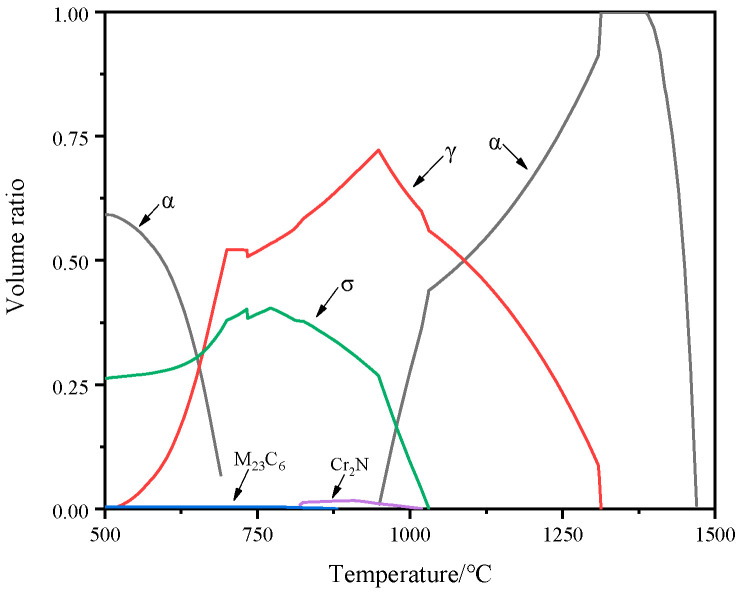
Thermodynamic calculation of the main precipitated phases and temperatures.

**Figure 3 materials-17-03390-f003:**
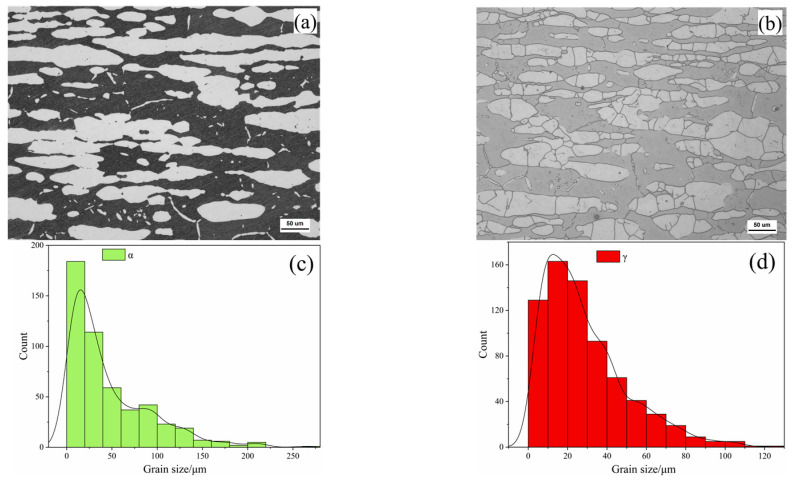
Two phase morphology and grain size: (**a**) lamellar two-phase morphology; (**b**) two-phase grain boundaries; (**c**) ferritic grain size distribution; (**d**) austenite grain size distribution.

**Figure 4 materials-17-03390-f004:**
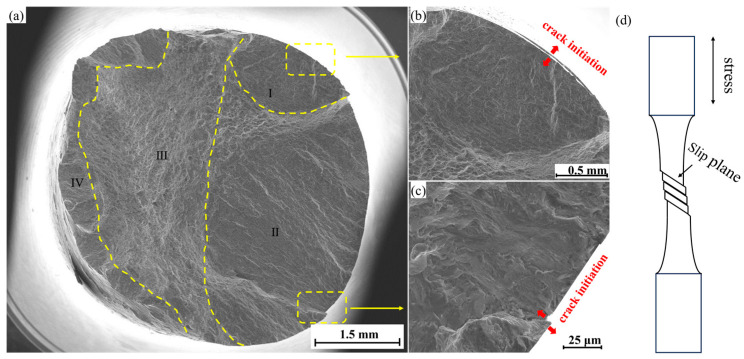
Fatigue fracture morphology and surface crack formation: (**a**) the macro-fracture morphology; (**b**,**c**) location of the crack source; (**d**) schematic diagram of fatigue fracture.

**Figure 5 materials-17-03390-f005:**
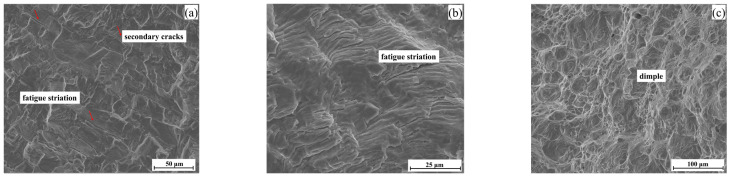
The micro-morphology of fatigue crack propagation zone: (**a**) enlarged of zone I; (**b**) enlarged of zone II; (**c**) enlarged of zone III.

**Figure 6 materials-17-03390-f006:**
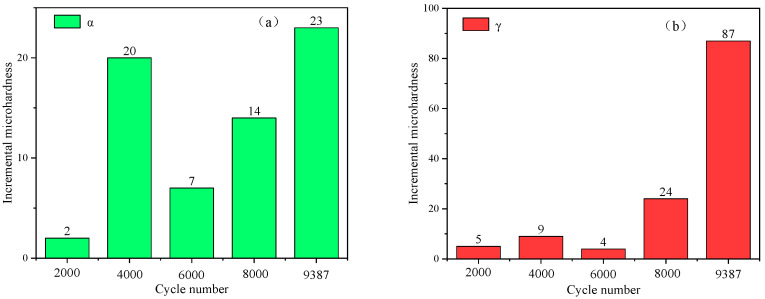
Two-phase microhardness increment under different cycle cycles: (**a**) ferrite; (**b**) austenite.

**Figure 7 materials-17-03390-f007:**
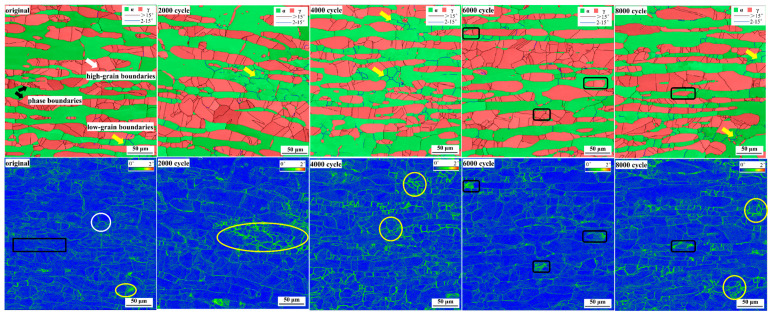
Two-phase grain boundary distribution and KAM distribution from the initial state to 8000 cycles.

**Figure 8 materials-17-03390-f008:**
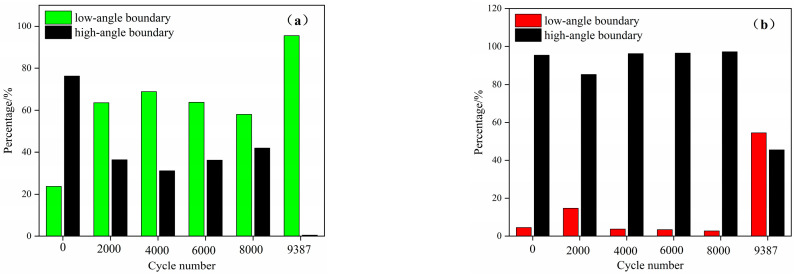
Grain boundary ratio from initial to fracture state: (**a**) ferrite; (**b**) austenite.

**Figure 9 materials-17-03390-f009:**
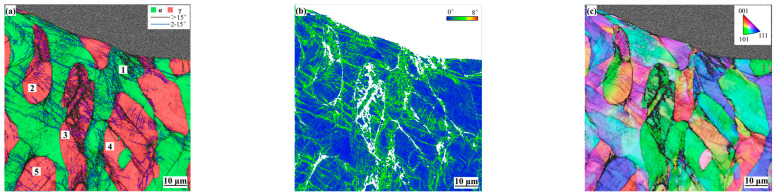
EBSD information of two-phase fracture at 9387 cycles: (**a**) grain boundary distribution; (**b**) KAM distribution; (**c**) orientation distribution.

**Figure 10 materials-17-03390-f010:**
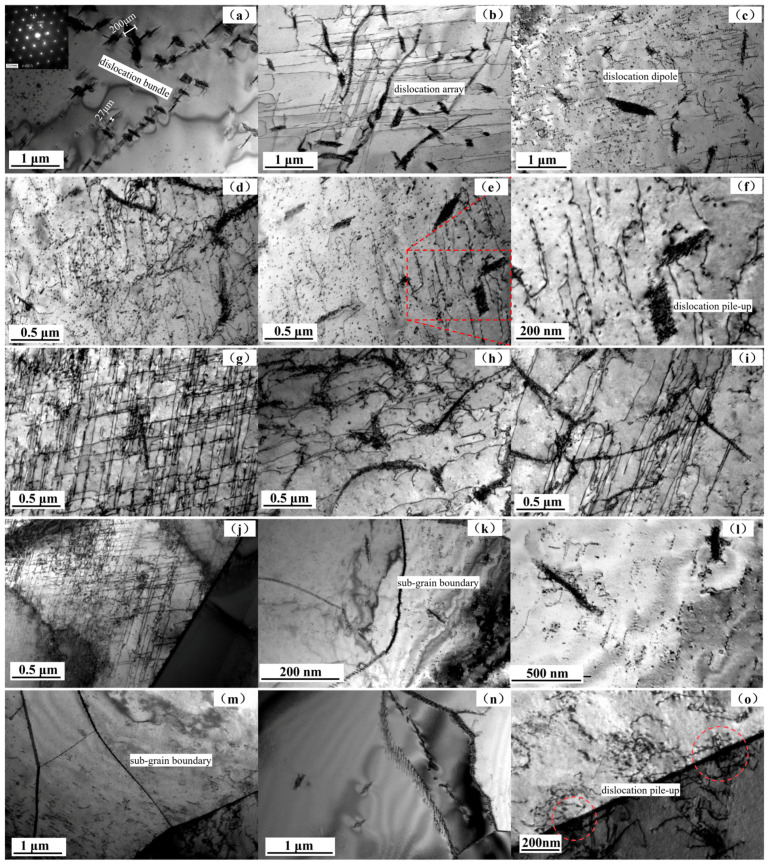
Dislocation morphology in ferrite grains: (**a**–**c**) initial state; (**d**–**f**) state of 2000 cycles; (**g**–**i**) state of 4000 cycles; (**j**–**l**) state of 6000 cycles; (**m**–**o**) state of 8000 cycles.

**Figure 11 materials-17-03390-f011:**
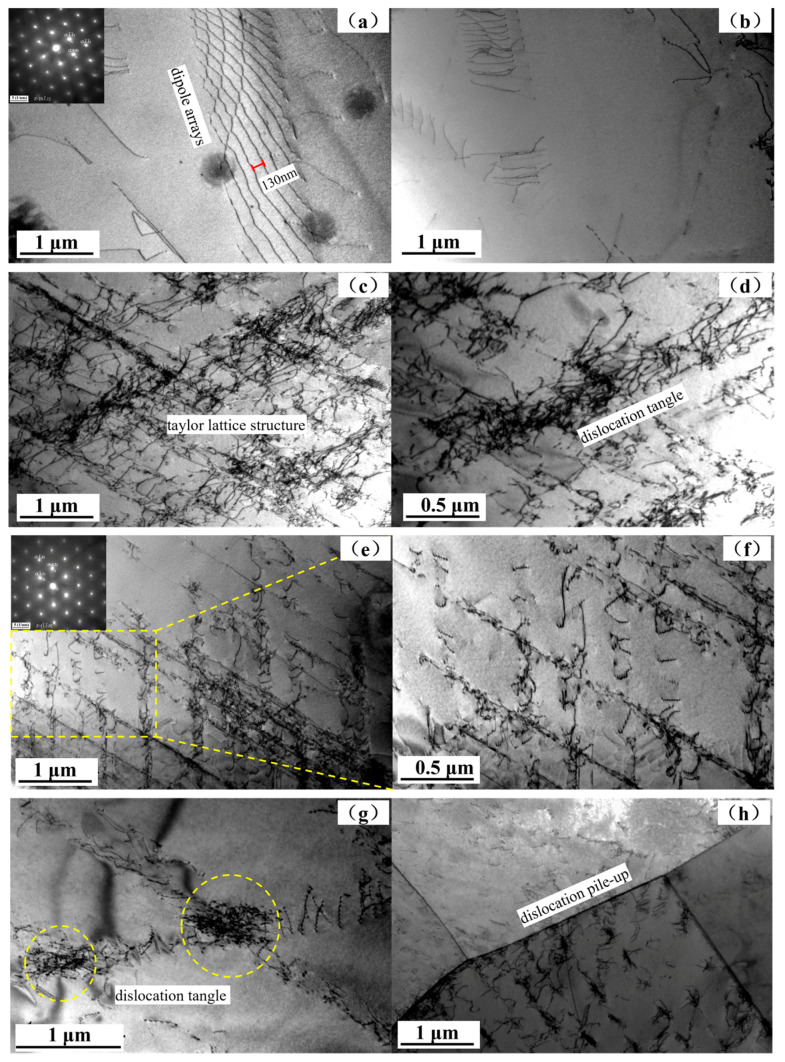
Dislocation morphology in austenite grain: (**a**,**b**) initial state; (**c**,**d**) states of 2000 cycles; (**e**,**f**) state of 4000 cycles; (**g**) state of 6000 cycles; (**h**) state of 8000 cycles.

**Figure 12 materials-17-03390-f012:**
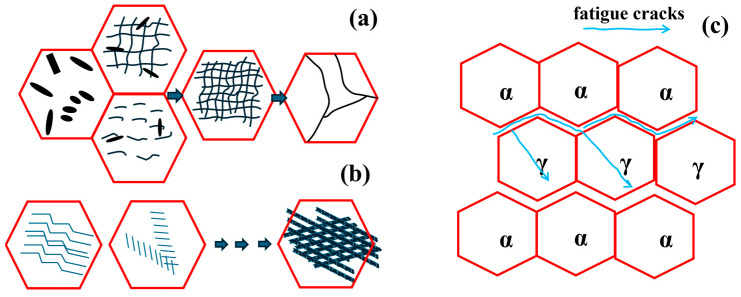
Ferrite phase dislocation evolution (**a**); austenite phase dislocation evolution (**b**); fatigue crack initiation and propagation (**c**).

**Table 1 materials-17-03390-t001:** Main chemical composition of test steel (mass fraction %).

C	Si	Mn	P	S	Cr	Ni	Cu	Mo	N	O
0.021	0.37	0.68	0.019	<0.0004	25.54	6.28	0.15	3.90	0.28	0.0022

**Table 2 materials-17-03390-t002:** Fatigue test parameters of the five samples.

Number	Maximum Stress/MPa	Stress Ratio (R)	Frequency/Hz	Cycle Number
1	600	−1	1	2000
2	600	−1	1	4000
3	600	−1	1	6000
4	600	−1	1	8000
5	600	−1	1	9387

## Data Availability

The original contributions presented in the study are included in the article, further inquiries can be directed to the corresponding author.

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
