# Peer review of "Study on the Deformation Behavior of Two Phases during the Low Cycle Fatigue of UNS S32750 Duplex Stainless Steel"

_materials, 2024, doi:10.3390/ma17143390_

Round 1

Reviewer 1 Report

Comments and Suggestions for Authors

The study has some scientific value, since it clarifies the mechanisms of fatigue failure of two-phase ferritic-austenitic steel in the low-cycle fatigue regime. The points in favor are good TEM studies of the dislocation structure of steel at different stages of fatigue loading. Based on the results of the study, the authors proposed a model of fatigue failure of two-phase steel.

The paper requires moderate revision before final publication.

1) How was such a low (<0.0004%) sulfur content controlled? And why is this sulfur content needed?

2) The maximum cycle stress is above the static yield strength of the steel. Why was such a high cycle stress used?

3) Line 107. Round to the first decimal place the proportion of ferrite and austenite.

4) Line 112. The authors provide the average grain sizes of ferrite and austenite. However, the histograms of grain size distribution do not correspond to a normal distribution. Therefore, there is no point in calculating the average value.

5) Line 128. ‘and’ twice in a row.

6) Line 134. What is exsit?

7) Line 138. Better ‘enlarged’ instead of ‘amplification’.

8) Line 150. There is no space after the decimal point.

9) The authors call zone III in the fracture one of the zones of fatigue crack propagation. However, this is obviously a static rupture, which does not belong to the fatigue crack propagation stage.

10) Explain the microhardness increments in Fig. 3-5. Were they calculated relative to the initial microhardness or relative to each subsequent loading stage? Otherwise, the nonmonotonic change in microhardness during loading is unclear.

11) Inscriptions in Figs. 3-2, 3-4, 3-6, and 3-7 should be increased.

12) Line 221. What is ‘road direction’?

13) The References List must be formatted in accordance with the requirements of the Journal.

Comments on the Quality of English Language

There are typos and unclear words/phrases.

Author Response

Dear chief editor:

    Thank you very much for taking the time to review this manuscript. The questions you raised show that you are a rigorous and professional scientific researcher. I have learned a lot, which has triggered me to think a lot. I should be more rigorous in scientific research in the future.  For each question you raised, I would like to discuss and learn from it with you, and the following is my reply.

Comment 1: How was such a low (<0.0004%) sulfur content controlled? And why is this sulfur content needed?

Response 1: Thank you for pointing this out. The UNSS32750 duplex stainless steel studied in this paper comes from Sandvik, Sweden, and its sulfur content is very low, and the sulfur content in China's smelting steel is about 25ppm. In the follow-up work, we also compare the properties of different sulfur content test steels at home and abroad.

Comment 2: The maximum cyclic stress is higher than the static yield strength of the steel. Why use such a high cyclic stress?

Response2:Thank you for pointing this out. When designing the experiment, we wanted to set the experimental conditions to be more demanding, so that it would be easier to explore the deformation behavior of stress and strain concentration and two-phase cycling.

Comment3:Line 107. Round to the first decimal place the proportion of ferrite and austenite.

Response3:I agree with this comment. I made changes in the manuscript.

Comment4: Line 112. The authors provide the average grain sizes of ferrite and austenite. However, the histograms of grain size distribution do not correspond to a normal distribution. Therefore, there is no point in calculating the average value.

Response4:Thank you for pointing this out. The description of average grain size has been deleted.

Comment5:Line 128. ‘and’ twice in a row.

Response5: Thank you for pointing this out. The superfluous and in the original text has been deleted.

Comment6:What is exsit?

Response6: Thank you for pointing this out. The spelling error has been corrected

Comment7: Better ‘enlarged’ instead of ‘amplification’.

Response7: I agree with this comment. I made changes in the manuscript.

Connent8: Line 150. There is no space after the decimal point.

Response8: Agree. I made changes in the manuscript.

Comment9:The authors call zone III in the fracture one of the zones of fatigue crack propagation. However, this is obviously a static rupture, which does not belong to the fatigue crack propagation stage.

Response9:Thank you for pointing this out. I've made changes in the manuscript.

Comment10:Explain the microhardness increments in Fig. 3-5. Were they calculated relative to the initial microhardness or relative to each subsequent loading stage? Otherwise, the nonmonotonic change in microhardness during loading is unclear.

Response10:  I agree with this comment. The microhardness increment is the microhardness under load stage minus the initial microhardness value.

Comment12: Line 221. What is ‘road direction’?

Response12: Thank you for pointing this out. The spelling error here has been corrected.

Comment13: The References List must be formatted in accordance with the requirements of the Journal.

Response13: Thank you for pointing this out. I have modified the reference format according to the published papers in the journal

Reviewer 2 Report

Comments and Suggestions for Authors

Check the file.

Comments on the Quality of English Language

Some terminology needs to be improved for the paper. The rest is fine.

Author Response

Dear reviewer:

    Thank you very much for taking time out of your busy schedule to review my manuscript. I agree with many of the comments you have given. Through these comments, I also know that you are a very rigorous and professional expert. I would like to discuss and learn with you through my reply. The following are the specific answers.

Comment1: S32750 is based on the UNS standard. Therefore, it should be correctly cited as UNS S32750.

Response1: I agree with this comment. I made some adjustments in the manuscript.

Comment2:  Typically, values are listed from low to high. The current format of 25 to 17% does not follow this convention. Please adjust accordingly.

Response2: Thank you for pointing this out. I have finished the revision.

Comment3:   Φ denotes diameter. Does Φ180 x 200 mm ingot mean 180 mm in diameter and 200 mm in height? Ensure this is accurately specified.

Response3: I agree with this comment. I have made clear marks in the manuscript.

Comment4: Refer to existing papers when writing your thesis. Although "Two-phase" is correct, existing literature often uses the term "dual phase."

Response4: Thank you for pointing this out. The two phases in the matrix are usually described as "two phases" or "both phases", and the "dual phase" is usually used to describe martensitic-ferrite dual phase steel. "duplex" is often used to describe duplex stainless steel.

Comment5: Add the experimental purpose to the conclusion. Adhering to basic formats is crucial for thesis publication. 

Response5: I agree with this comment. I added the corresponding details to the conclusion of the manuscript

Reviewer 3 Report

Comments and Suggestions for Authors

-        Please consider in the abstract practical application, innovation and scientific significance of your paper. What is new in your paper in comparison with other papers and authors? How you move step forward in scientific approach and methodology of analysed problem especially in comparison with approaches of other scientists?

-        In Introduction section, please make more detailed investigations of other papers and give more precisely review of scientific investigations regarding anaylsed topic. Please mention specifics of presented papers and position your paper appropriately to emphasize your scientific contribution.

-        In section 2.1. please show figures of experimental specimens.

-        In section 2.2. please give table of experimental setup with parameters that were kept constant as well as with those that were varies in experimental trials.

-        Please add in section 2 flow diagram that will present your experimental work more clearely.

-        Please add in section 2 figures of experimental equipment and maesurements machines.

-        Please  fill section 2 with sketches that will present measurements setup and positions.

-        Please, numerate figures as: Fig.1, Fig.2, Fig.3 …

-        Please modify section 4.3 with figures and explanations for your own specimens.

-        Please determine in conclusion practical view of conducted research, advantages and disatvantages as well as potential topics for future investigations.

Author Response

Comment 1: Please consider in the abstract practical application, innovation and scientific significance of your paper. What is new in your paper in comparison with other papers and authors? How you move step forward in scientific approach and methodology of analysed problem especially in comparison with approaches of other scientists?

Response 1: Thank you for pointing this out. I'm not innovative in my approach. I think my innovation is to find that cracks are easy to produce in the phase boundary and in the austenite, and to summarize the evolution of the two-phase dislocation structure during the cycle.

Comment 2: In Introduction section, please make more detailed investigations of other papers and give more precisely review of scientific investigations regarding anaylsed topic. Please mention specifics of presented papers and position your paper appropriately to emphasize your scientific contribution.

Response 2: Yes, I agree. I have revised the introduction according to your comments. Specific changes are noted in the manuscript.

Comment 3:  In section 2.1. please show figures of experimental specimens.

Response 3: Thanks for pointing this out. Chapter 2.1 is mainly about the preparation of materials, including the forging and rolling of steel ingot. Do you want me to supplement the slab pictures after forging and rolling?

Comment 4: In section 2.2. please give table of experimental setup with parameters that were kept constant as well as with those that were varies in experimental trials.

Respone 4: Yes, I agree. I have added to the manuscript a table describing the parameters of the experiment. Specific changes have been noted in the manuscript.

Comment 5: Please, numerate figures as: Fig.1, Fig.2, Fig.3 …

Response 5: Thank you for pointing this out. I have completed the modification according to your suggestions. Specific changes have been noted in the manuscript.

Comment 6: Please determine in conclusion practical view of conducted research, advantages and disatvantages as well as potential topics for future investigations.

Response 6: Yes, I agree. I made changes in the manuscript and made notes.

Round 2

Reviewer 1 Report

Comments and Suggestions for Authors

It can be accepted

Author Response

Dear reviewer:

    Thank you for taking time out of your busy schedule to review my manuscript, and I am very happy to get your approval.

I wish you all the best.

Reviewer 2 Report

Comments and Suggestions for Authors

The paper has been overall improved.

It has been revised to align with the journal format.

However, some areas still need improvement:

1) The paper transitions directly from section 3.2 Mechanical Characterization to section 5. Conclusion. This needs to be addressed.

Example: Add a section 4. Discussion or renumber the conclusion as section 4.

2) The first sentence of the Conclusion should indicate what the conclusion is based on.

3) Captions throughout the paper need to be enhanced (e.g., Figure 9).

Once these three points are addressed, I support the publication of this paper.

Author Response

Dear reviewer:

    Thanks again for taking time out of your busy schedule to review my paper.I really appreciate your advice, at the same time I finished the revision.Every change is marked in the manuscript.The following is my specific reply to your comments.

Comment 1: The paper transitions directly from section 3.2 Mechanical Characterization to section 5. Conclusion. This needs to be addressed.

Response 1: Thank you for your comment. I have divided the third section into two section.

Comment 2: The first sentence of the Conclusion should indicate what the conclusion is based on.

Response 2: I agree with this opinion. I add a theoretical basis to the first conclusion

Comment 3: Captions throughout the paper need to be enhanced (e.g., Figure 9).

Response 3: Thank you for your comment. I have rechecked and revised the titles of the pictures in the manuscript.

Reviewer 3 Report

Comments and Suggestions for Authors

- Please supplement Comment 1 and Comment 3 (figures) from my first review

Comments on the Quality of English Language

- Minor editing is required

Author Response

Comment 1: The introduction including the description of the state of the art and the research objectives is very short and should be extended to cover all relevant papers in the field and related to the analysis carried out by the authors. Those that are already mentioned should be discussed in more detail. For instance, a discussion of the dislocation structures occuring during fatigue is completely missing.

Response 1: Thank you for pointing this out. I actually changed the introduction in the last revision. Perhaps the previous cover letter was not clear enough on this point. I have reorganized and revised the introduction to introduce the research results and analysis in related fields in more detail, and added the relevant research results on the evolution of dislocation structure during fatigue.

Comment 3: “Duplex stainless steel (DSS) is a kind of stainless steel with similar volume fraction of α phase and γ phase, which good corrosion resistance and excellent mechanical properties are well combined.” => “Duplex stainless steel (DSS) is a kind of stainless steel with similar volume fraction of α phase and γ phase, which combines good corrosion resistance and excellent mechanical properties.”

Response 3: Yes, I agree. In fact, I revised this sentence in accordance with your comments in the last manuscript and marked it in the manuscript. I noticed that I left out a reply to this point in my last reply letter.
